# Immersive virtual reality-based intervention for psychological wellbeing among older adults: A systematic review and meta-analysis

Jing Jing Su[1,2], Chi-Keung Chan[2,3], Ladislav Batalik[4,5,6,7], Wai Chung Chung[8], Chen Lei[1], Rick Yiu Cho Kwan[1,2]*

1 School of Nursing, Tung Wah College, Hong Kong SAR, China, 2 Translational Research Centre for Digital Mental Health, Tung Wah College, Hong Kong SAR, China, 3 School of Arts and Humanities, Tung Wah College, Hong Kong SAR, China, 4 Department of Rehabilitation, University Hospital Brno, Brno, Czech Republic, 5 Rehabilitation Clinic, Faculty of Medicine, Masaryk University, Brno, Czech Republic, 6 Department of Physiotherapy and Rehabilitation, Faculty of Medicine, Masaryk University, Brno, Czech Republic, 7 Department of Public Health, Faculty of Medicine, Masaryk University, Brno, Czech Republic, 8 Department of Special Education & Counselling, The Education University of Hong Kong, Hong Kong SAR, China

* rickkwan@twc.edu.hk

## Abstract

Immersive virtual reality (IVR) is an emerging therapeutic modality that engages older adults in psychological therapeutically oriented activities developed to improve their psychological well-being. This systematic review aims to investigate the effects of IVR psychological intervention on psychological symptoms and well-being. A systematic review and meta-analysis was conducted following the Cochrane Handbook for Systematic Reviews of Interventions. Six databases were searched, including Embase, PubMed, Web of Science, Scopus, CINAHL, and PsycINFO, covering the period from 2010 to December 2024. RevMan 5.3 was utilized for meta-analysis, and the Cochrane Risk of Bias tool was employed for quality assessment. Ten randomized controlled trials of 746 older adults were included. The IVR interventions employed reminiscence (40%), garden/forest therapy (40%), cognitive stimulation (10%), and multi-sensory stimulation to reduce psychological symptoms and improve self-perception (10%). Data pooling suggested that IVR interventions have significantly reduced depressive symptoms [n = 5; SMD = -0.83, 95%CI (-1.05, -0.60), $I^2$ = 21%, $p$ < .001]; anxiety [n = 5, SMD = -0.77, 95% CI (-1.32, -0.22), $I^2$ = 70%, $p$ = .006]. Synthesis without meta-analysis (SWiM) was conducted for stress and affect outcomes following SWiM guidance. In all three studies (100%), IVR produced statistically significant reductions in stress versus usual/standard care, and in both studies (100%), it yielded statistically significant improvements in affect—higher positive and lower negative affect—compared with the respective control conditions. IVR-based interventions could be an alternative method for alleviating the psychological symptoms of older adults.

**Registration:** PROSPERO CRD42024575387

**Data availability statement:** All relevant data are within the manuscript and its Supporting information files.

**Funding:** This research was funded by the Ministry of Health, Czech Republic, conceptual development of research organization (FNBr, 65269705 to LB). This research was supported by a grant from the Research Grants Council of the Hong Kong Special Administrative Region, China (Project No., UGC/IDS(R)17/23 to CKC). The funders had no role in study design, data collection and analysis, decision to publish, or preparation of the manuscript.

**Competing interests:** The authors have declared that no competing interests exist.

## Author summary

This study investigated the effects of immersive virtual reality (IVR)-based psychological interventions in alleviating psychological symptoms and enhancing well-being among older adults. The novelty of this research lies in its pooling of results from updated randomized controlled trials that exclusively examine the immersive format of VR-delivered psychological interventions. Previous studies often combined varying levels of VR immersiveness or mixed different types of IVR-based interventions (e.g., exercise, cognitive assessment, psychological), resulting in high heterogeneity that impeded definitive conclusions. The impact of this study is demonstrated by employing advanced immersive technology to create vivid and interactive experiences (e.g., garden environments) that achieve therapeutic effects. The meta-analysis results revealed that IVR-based psychological interventions significantly reduced depressive symptoms and anxiety among older adults, indicating the high potential of this modality for improving psychological outcomes, particularly for those with mobility limitations.

## Introduction

As populations age worldwide, older adults experience a range of mental health challenges [1,2]. Beyond predominant concern of anxiety and depression, these include loneliness and social isolation, chronic stress, and the psychological burden of multimorbidity and intrinsic capacity decline [1–3]. In response, the World Health Organization (WHO) calls for reorienting health systems toward healthy aging and expanding access to evidence-based psychological interventions to prevent and alleviate later-life mental health problems [4]. Likewise, the United Nations' Madrid International Plan of Action on Aging highlights that the prevalence and impact of mental health conditions rise with age and that co-occurrence is common [5]. The simultaneous presence of mental disorders frequently leads to reduced treatment efficacy and can contribute to notable psychological deterioration [6]. Therefore, strategies to promote active, healthy aging should encourage interventions that alleviate psychological symptoms, strengthen overall well-being, social connectedness, and cognitive and emotional resilience.

Psychological intervention is a structured, non-pharmacological procedure that employs psychological principles to modify a person's emotions, attitudes, or behaviors through processes such as education, training, therapy, or support [7]. Immersive virtual reality (IVR), a computer-based technology, is increasingly used in contemporary psychological interventions to enable older adults' participation in structured psychotherapeutic activities [8,9]. IVR is featured by its capacity to generate a compelling sense of physical presence within a computer-generated environment. This spatial immersion is achieved through head-mounted displays and related peripherals that envelop the user in coherent visual, auditory, and sensory stimuli, rendering the virtual setting perceptually authentic [10]. This differs from non-immersive VR,

which utilizes devices like a monitor, keyboard, and mouse to display a 3D environment without enclosing the user to produce the same feeling of "being there." It is this immersive quality that unlocks the full potential of VR technology in delivering psychological intervention [10,11]. By offering a multisensory, immersive environment that simulates real-world scenarios, IVR can engage older adults emotionally with minimal distractions [12–14]. This technology fosters a strong sense of presence and embodiment within the virtual environment, reportedly enhancing emotional responses among older adults [15]. Recently, a growing number of randomized controlled trials (RCTs) have incorporated IVR interventions to improve the psychosocial well-being of older adults [16]. IVR provides a highly adaptable environment accommodating diverse needs, disabilities, and goals, offering variations in preferences and morbidity levels [17]. By immersing participants in expansive environments representing real-life scenarios, IVR provides safe, accessible, and engaging methods for therapeutic engagement.

There is increasing evidence regarding the effect of IVR interventions in improving the health outcomes of older adults, with or without cognitive impairment [15,18]. Two review studies have explored the application of IVR-based interventions for the health and well-being of older adults residing in long-term care facilities [18] and both residential aged care facilities and community settings [15]. Through narrative synthesis, these studies concluded that IVR is a promising, safe, and enjoyable modality for promoting the health and well-being of older adults [15,18]. However, the current uncertainty regarding the effectiveness of IVR-based interventions necessitates more robust evidence. Another systematic review and meta-analysis included IVR based psychological and physical interventions, as well as cognitive assessment tools studies, suggested a positive impact in reducing anxiety, depression, and enhancing psychological well-being [19]. However, existing reviews have included various types of IVR-based interventions (e.g., exercise, cognitive, psychological), which may generalize the research questions and restrict the validity of their conclusions and implications. Additionally, there is limited understanding of commonly used strategies for delivering IVR interventions to older adults. Moreover, it's implementation faces practical challenges such as cybersickness and visual discomfort; accessibility for individuals with sensory or cognitive impairments; the need for staff training, supervision, and dedicated space; and device cost, maintenance, and technical support—all of which can affect adherence, safety, and effectiveness [20,21].

Importantly, prior systematic review with meta-analysis pooled IVR interventions across exercise, cognitive training, and psychological modalities, approaches that engage different mechanisms and target distinct outcomes, potentially diluting mental health effect estimates and obscuring best evidence generation. Psychological IVR interventions warrant isolated focus because they uniquely leverage immersion and presence, stimulating emotional, cognitive, and behavioral processes that influence anxiety, depression, affect, and subjective well-being; moreover, they require distinct dosing, facilitation, and IVR content compared with non-psychological IVR [22]. Therefore, this systematic review and meta-analysis aim to investigate the effects of IVR psychological interventions on the psychological wellbeing of older adults.

## Methods

This review was guided by the Cochrane Handbook for Systematic Reviews of Interventions (Cochrane Collaboration, Oxford, UK) and presented using the Preferred Reporting Items for Systematic Reviews and Meta-Analyses (PRISMA) statement [23]. The systematic review protocol was registered in PROSPERO (CRD42024575387).

### Eligibility criteria

The Populations, Interventions, Comparisons, Outcomes, and Study designs (PICOS) framework was used to guide eligibility criteria. The inclusion criteria were: (1) P: older adults with or without medical conditions; (2) I: the intervention group received immersive virtual reality interventions for psychosocial wellbeing (e.g., cognitive behavioral therapy, reminiscence, stimulation activities, relaxation therapy, mindfulness); (3) C: any form of control, including but not limit to usual care, waitlist, placebo, or online education, (4) O: results measured regrading psychological symptoms (e.g., anxiety, depression), affect (e.g., mood), psychosocial wellbeing; and (5) S: randomized controlled trials (to generate the best evidence) [24].

The exclusion criteria were as follows: (1) quasi-experimental, qualitative, or case studies; (2) studies without attempting to immerse the participants inside the technology generated world or step inside the synthetic world (non-immersive mode), because our focus is on IVR-specific mechanisms—such as sense of presence and multisensory immersion—that are theorized to drive psychological outcomes; (3) studies that did not involve older persons themselves, but their caregivers; (4) conference abstracts; and (5) unavailable full-text even after contacting the authors.

## Search methods

A comprehensive search was conducted in six databases, including Embase, PubMed, Web of Science, Scopus, CINAHL, and PsycINFO, covering the period from 2010 to December 2024, using the PICOS framework. The initiation time was set to 2010, during which low cost immersive virtual reality technology became commercially available [25]. Medical Subject Headings (MeSH) terms were used and presented in S1 File.

## Study selection

Endnote (version 21; Clarivate) was used to remove duplicates, and two teams (team A: JJS, LB, WCC, team B: RK, CL, CKC) independently reviewed studies by judging titles and abstracts against predefined criteria. Studies that fulfill the inclusion criteria or judged uncertain progressed to full-text review. Two teams conducted full-text review independently, and any disagreement on study inclusion was resolved by team discussions immediately after screening to ensure objectivity and vigor.

## Study quality and risk of bias

Methodological quality was assessed independently by two researchers (WCC and CL) using the Cochrane Risk of Bias tool (version 2), and any disagreement in interpretation of data was resolved by a third author (JJS). Study quality was assessed regarding selection bias, performance bias, attrition bias, detection bias, reporting bias, and intention to treat analysis [26]. Risk of bias was judged as unclear, low, or high when data were insufficient or uncertain.

## Data extraction

All study data were extracted independently by two authors (CL &WCC) and any disagreement in interpretation of data was resolved by a third author (JJS). The authors developed a table to extract data, including: (a) origin of the articles: authors, year, and country; (b) sample characteristics: sample size, setting, age, gender, and diagnosis; (c): group design: brief description of intervention and control group, (d) assessment time point, (e) health outcomes and instruments, and (f) attrition rate. Details of the interventions were extracted in concordance with the TIDieR checklist [27].

## Data analysis

Review Manager 5.3 (Nordic Cochrane Centre, Cochrane Collaboration) was used for data pooling when three or more studies were reporting the same outcome; otherwise, synthesis without meta-analysis (SWiM) was conducted following SWiM guideline [28]. Where the same study reported outcomes at different endpoints, data from post-intervention were used to investigate the effectiveness. For studies using cross-over design, the first post-test results were pooled to improve consistency across studies. For each outcome of interest, an intervention effect size was expressed as Hedges' adjusted g [29], calculating standard mean difference (SMD) with a 95% confidence interval (CI) of post-intervention results between groups. Effect size >0.8 describes a large effect, 0.5–0.8 a medium effect, and 0.2–0.5 a small effect [30]. Statistical heterogeneity between studies was quantified with the $I^2$ statistic. The SMD was computed using a random effect model for $I^2 > 50\%$ [31]. Leave-one-out sensitivity analysis was conducted when the pooled effect showed significant heterogeneity. The certainty of evidence was assessed using the Grading of Recommendations Assessment, Development and

Evaluation (GRADE) approach. Under GRADE, RCTs start as high-certainty evidence and are downgraded for risk of bias, inconsistency, indirectness, imprecision, or publication bias [32]. Tests for funnel plot asymmetry were not undertaken in this meta-analysis because fewer than ten studies were pooled, a number too small to reliably distinguish random variation from true asymmetry [33]. Instead, publication bias was evaluated and reported in terms of selective outcome reporting by cross-referencing the included studies with their trial registrations and/or published protocols.

## Results

The PRISMA flow diagram illustrating the study selection process is presented in Fig 1. Initially, 3,986 studies were identified through the literature search. After removing 686 duplicates and screening 3,300 titles and abstracts, 32 articles remained. Following a full-text review, 10 RCT studies met the eligibility criteria, as detailed in Table 1 [34–43]. The search strategies for each database are provided as a supplementary file (S1 File).

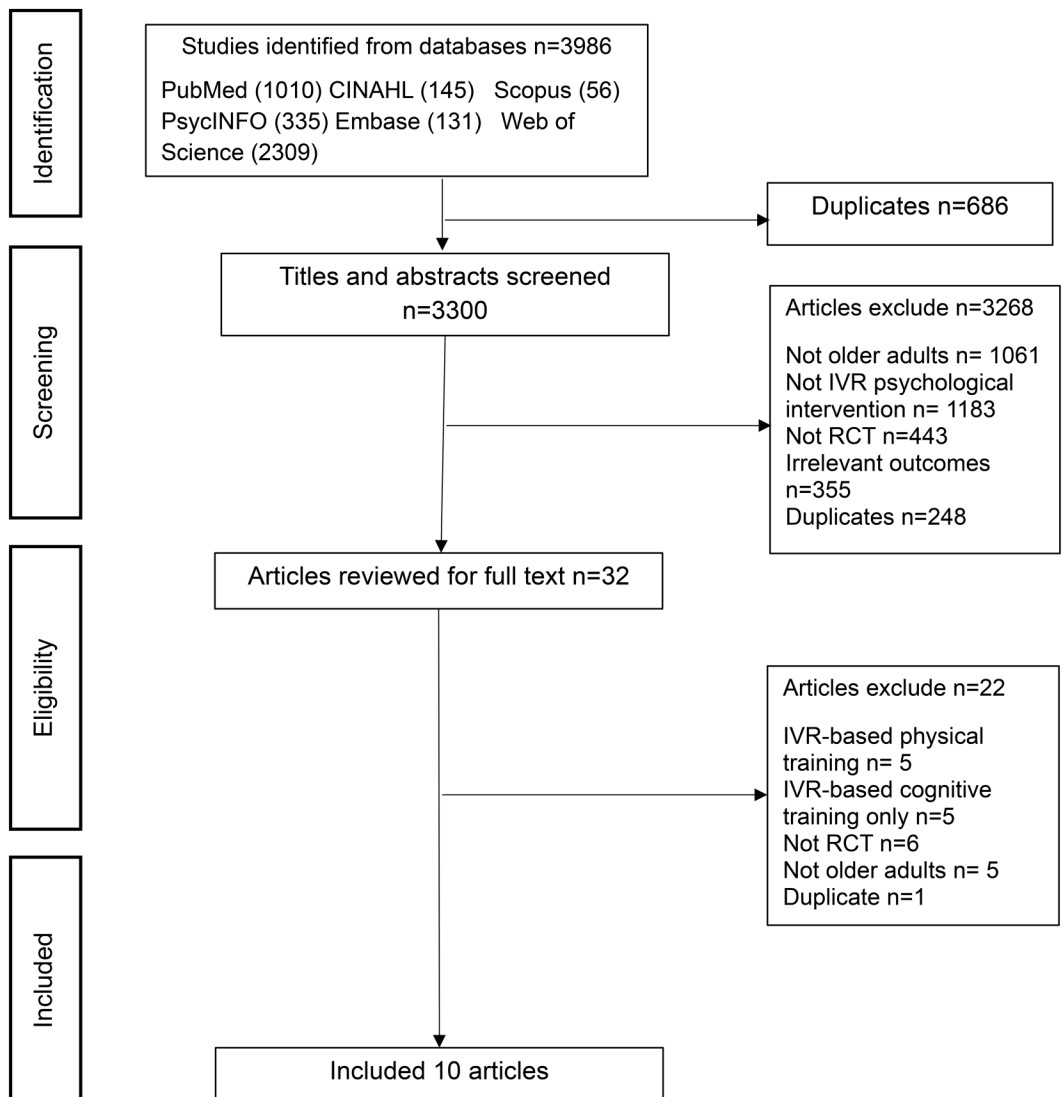

**Fig 1. PRISMA diagram of identifying studies of IVR.** n represents the number of studies.

**Table 1. Characteristics of the reviewed studies (PICOT).**

| Population | | | | Intervention | | | Control | Outcome | Timeframe | Remarks | |
|---|---|---|---|---|---|---|---|---|---|---|---|
| First author, year, country/ region | People & recruitment setting | No. of participants (total/ IG/ CG) | Mean age (SD) & gender ratio (female) | Intervention setting & duration | Content | Dosage | | Instruments | Assessment point | Attrition | Adverse event of IVR group |
| Abd El Fatah, 2024 Egypt | Participants from 4 elderly homes: a) ≥60 years old b) Communicate clearly c) Tolerate head mounted display | 60, IG: 20 Reminiscence group: 20 CG: 20 | IG: 65.95 (3.66) Reminiscence group: 66.80 (4.65) CG: 67.30 (4.41); 32 (53%) | Activity room of the elderly centre, 6 months | IG: IVR reminiscence Traditional reminiscence | 30 to 45 -min/d, 2d/ week, 12 sessions | Usual care | Revised Riff's Psychological Well-Being Scale | Baseline, one week & 3-month after intervention | 6/66 (9.1%) | Very mild nausea, vestibular/ oculomotor disturbance reported by 5–20% of participants |
| Niki, 2020 Japan | ≥75 years old who can recognize VR images were recruited from single nursing home. | 10, IG: 5 CG: 5 | IG: 84.4 (2.3) CG: 89.8 (4.0); 6 (60%) | Nursing home, 1 week | Live action (LA) IVR versus Computer graphics (CG) IVR | 10-min/ round, 2 rounds/ day | Computer graphics | State-Trait Anxiety Inventory (STAI) | Baseline, after each round of VR sessions | No, all completed the study | Nausea: n = 1 Tiredness n = 2 Headache n = 1 |
| Kiper, 2022 Poland | Participants from rehabilitation centre who: a) 55–75 years old b) With ischemic stroke c) ≥10 in Geriatric Depression Scale (GDS)-30 | 60, IG: 30 CG: 30 | IG: 65.50 (6.72) CG:65.57 (4.99) 30 (50%) | Rehabilitation centre, 6 weeks | IG: IVR therapeutic garden & psychotherapy & upper extremities activity | 20-min/d, 10 sessions in 3 weeks | Schultz's Autogenic Training | Depressive symptoms: GDS-30 Hospital Anxiety and Depression Scale (HADS) | Baseline, 3 weeks, 6 weeks | 21/60 (35%) | No adverse events were reported |
| Chan, 2020 Hong Kong | Participants from community center: a) ≥ 60 years old b) Read and speak Cantonese c) Normal vision with/ without glasses | 235; Group A: 129 Group B: 106 | IG: 72.7 (7.78) CG: 75.0 (8.13). 153 (65.11%) | Elderly center, 2 weeks | IVR scenery of HK in the past 20 yrs | One 20–25 min VR interaction | Paper-and-pencil (PP) cognitive stimulation | Affect: Positive and Negative Affect Score (PANAS) | Baseline, After IVR activity, and after paper-and-pencil activity | 59/236 (25%) | 3 participants (1.4%) reported severe simulator sickness |
| Brito, 2021 Chile | Participants Recruited through door-to-door who: a) 55–90 years old b) With mild cognitive impairment | 113; IG: 60 CG: 53 | Overall mean = 66.90 (SD = 7.14). 53 (46.90%) | Laboratory setting. 6 weeks | IG: IVR-based sensorimotor rehabilitation 4 types of virtual environment | 3 times a week with 25 min per session | Usual care | RYFF Well-being scales Depression: YESAVAFE scale Anxiety: General Health Questionnaire | Baseline, Immediate post-intervention measurement | 27/140 (19.25%) | NA |

*(Continued)*

| Population | | | | Intervention | | | Control | Outcome | Timeframe | Remarks | |
|---|---|---|---|---|---|---|---|---|---|---|---|
| Cieslik, 2023 Wroclaw, Poland | Recruited from Foundation for Senior Citizen Activation A) Age 60–85 B) 30 item GDS score <10 C) HADS score <8 | 60; IG: 30 CG:30 | IG: 68.77 (5.57) CG: 67.53 (5.51) Overall: 68.15 (5.53). 60 (100%) | Clinical setting, 4 weeks | IG: IVR-based relaxation sessions | 8 hourly sessions (40 min GFT session, 20 min IVR), 2/ wk | Group relaxation and psychoeducation | Geriatric depression scale (GDS) HADS | Measures at two time points: pre- and post-intervention | 0, all participants completed | NA |
| Mazurek, 2023 Poland | Recruited from hospital who: • ≥60 years old • Recently undergone hip or knee joint arthroplasty surgeries | 65; IG: 33 CG: 32 | IG: 69.71 (6.82) CG: 69.47 (5.52) Overall=69.59 (6.16). 42 (61.76%) | In hospitals, 4 weeks | IVR therapy (nature & Erickson's psychotherapy) calming music, mandala coloring, & mood-enhancing garden aesthetic | 8 VR therapy sessions, twice a week, 20 min each | Standard care | HADS Perceived stress scale (PSS-10) The Perception of Stress Questionnaire (PSQ) | Before and after intervention | 3/68 (4.42%) | NA |
| Szcze pańska-Gieracha, 2021 Wroclaw, Poland | Recruited from local clinical lab, which provides therapeutic program for older women: a) Age over 60 b) GDS score ≥ 10, or HADS-A≥ 8, or HADS-D ≥ 8. | 25 elderly women. IG:13 CG:12 | IG: 70.18 (4.87) CG: 71.25 (4.41) Overall: 70.73 (4.56) 25 (100%) | 4 weeks | Standard care+IVR garden | 8 VR session, twice a week | Standard care & 20 min health education & psycho-education | GDS-30 The Perception of Stress Questionnaire HADS | Baseline Post intervention Two weeks after the intervention | 2/25 (8%) | NA |
| Tominari, 2021 Japan | Recruited from four elder-care facilities a) Japanese speaking, aged 65 years or older b) Diagnosed with dementia No psychological disorders/ visual nor auditory impairments | 52 IG:26 CG:26 | IG=85.1 (range=69–98) CG=87.0 (range=68–98) 40 (76.92%) | 8 weeks | IVR Reminiscence therapy | One to one, 30–35 min per session, weekly | Conventional photo-based reminiscence therapy | Revised PGC morale scale Multidimensional Observation Scale for Elderly Subjects (MOSES) | Two endpoints: Baseline and post intervention | 2/54 (3.70%) | NA |
| Yuan, 2022 China | Recruited from elderly care institution Unspecified inclusion criteria | 63; IG=32 CG=31 | Overall: 81.98 (7.15) 42 (66.67%) | 3 consecutive days | IVR forest | 5 min IVR forest with 3 min of reliving | Usual care | Affect: Short version of PANAS Stress: Six-item restorative outcome scale | Two time points: Pretest and post-test | Not reported | NA |

## Study quality

Overall, studies had a low or medium risk of bias. Most were judged to have an unclear risk due to an inadequate report of allocation processes. Participants and personnel were not blinded across all studies, except one study participants were not aware of the study group assignments [39]. Three studies omitted a detailed description of allocation and blinding, which yielded unclear risks [36,37,43]. One study was judged to have a high risk of bias due to only 10 participants, and a significant age difference between the groups at baseline [35], as shown in Figs 2 and 3.

## Study characteristics

The ten included studies were published between 2020 to December 2024. All studies adopted a two-arm design, except one that used a three-arm design [34]. Four studies were conducted in Poland [38,41–43], two in Japan [35,36], one each in Chile [40], Hong Kong, China [39], mainland China [37], and Egypt [34]. A total of 746 participants were included in this review with sample size ranging from 10 [35] to 235 [39]. The attrition rate varied from 0% [41] to 35% [38]. The majority of participants were female (n = 483, 64.75%), with a mean age ranging from 65.5 ± 6.72 to 89.8 ± 4.0 years. The included studies recruited older adults from residential home [34–37], patients undergoing rehabilitation [38,42], or community [39–41,43]. Moreover, three studies specified that participants should also have depressive symptoms [38,43] or dementia [36]. The control group received standard usual care [34,37,40,42,43] or active control. The intervention ranged from one week to six months (Table 1).

## Study interventions

Details of the interventions are presented in Table 2 in concordance with the TIDieR checklist [27]. Four studies (40%) focused on IVR-based reminiscence [34–36,38], exploring themes such as city's hotspots' scenery in the past 20 years, childhood memory. Four studies employed natural exposure therapy (i.e., garden/forest) [37,41–43]. One study provided cognitive stimulation [39], whereas another implemented multi-sensory stimulation via four IVR scenarios to mitigate psychological symptoms and enhance self-perception [40]. Four studies (40%) reported using theory to guide their intervention, including Erickson's psychotherapy [38,41–43] and positive psychology [43]. The immersiveness of virtual reality intervention was facilitated using various devices: head mounted devices [34,35,37,40], VR goggles and controller [38,41–43], a wearable VR device [39], and a 360-degree panoramic displays [36].

Four studies (40%) incorporated professional input or validation regarding the IVR intervention: one study involved a certified psychotherapist to validate the VR therapy content [38,43], another study integrated music composed through

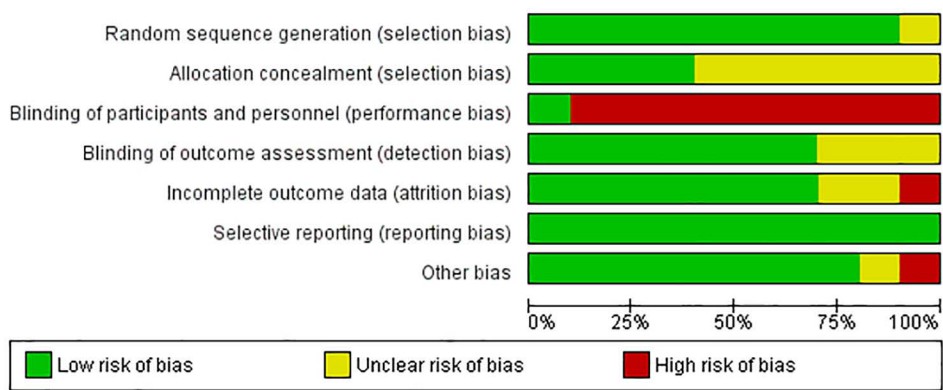

**Fig 2. Risk of bias summary.**

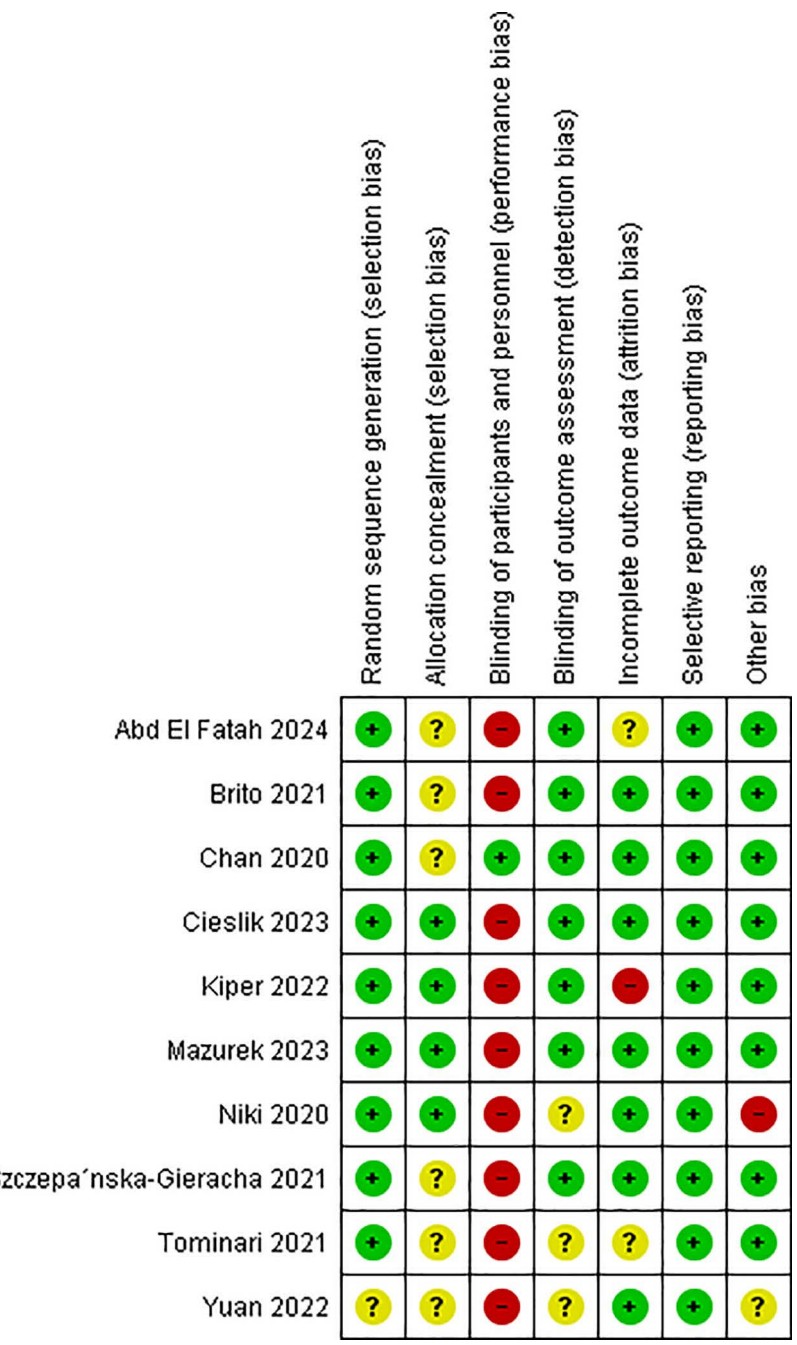

**Fig 3. Risk of bias graph.**

collaboration between a music therapist and a composer [41], and one study employed a certificated nurse trained in reminiscence therapy conducted all eight reminiscence sessions [36].

A series of strategies were employed to tailor the intervention. For example, image content was obtained by conducting reminiscence interviews with each participant to allow personalization in reminiscence therapy [34]. Other studies used live-action images captured by a 360-degree camera at a theme park and computer-generated images [35],

**Table 2. Description of the intervention following TIDieR checklist.**

| First author, year | Brief name | Goals & theory | Material & procedure | Professional input | how much | Tailoring | Actual usage |
|---|---|---|---|---|---|---|---|
| Abd El Fatah, 2024 | IVR reminiscence | Improve psychological well-being & cognitive performance among older adults in assisted living facilities. Theory: NA | VR head-mounted device VR Wander application (allow participants to explore their old town via Google Street) | NA | 12 sessions in 6 weeks, two days per week | Interviews were conducted to create materials (VR images/videos) based on participant's background | All completed 12 sessions |
| Niki, 2020 | The efficacy and safety of IVR reminiscence | Examined the efficacy and safety of IVR reminiscence on anxiety Theory: NA | Tablets for mirroring VR images VR headset 54 photos of Japan city view as live images Software FUNSET to edit CG photos | NA | 2 sessions (10–15 min each) in one day. | VR images were tailored with photos of a theme park in Fukuyama City to facilitate reminiscence. | All completed |
| Kiper, 2022 | VR therapy in supporting the recovery of depressive symptoms in post-stroke rehabilitation | Effectiveness of IVR therapy on both functional activity and depressive symptoms in stroke survivors. Theory: Erickson's psychotherapy | VR devices (HTC VIVE PRO Goggles and a controller) VR TierOne software presents a virtual therapeutic garden Headphones to facilitate SAT | The content of the VR therapy was developed by a certified psychotherapist of the European Association of Psychotherapy | 10 sessions in 3 weeks | Motor function training personalized to the individual patient's motor capacity, with a progressive increase in complexity | NA |
| Chan, 2020 | Cognitive stimulation VR activity | To investigate mood change and potential adverse events after IVR intervention Theory: NA | wearable VR devices, 5 VR scripts about Hong Kong's famous tour hotspots' scenery | NA | Single session in one day, with 5 VR scripts showed to participants | Tailor-made IVR scripts about tour spot in Hong Kong to stimulate memory recall | 129 participants experience VR activity |
| Brito, 2021 | IVR-based multi-sensory stimulation effect on mental health | Theory:<br>• IVR anxiety models (risky situations)<br>• Environmental exposure models<br>• Self-identification models (self-compassion) | Virtual reality helmet HTC vive Four IVR scenarios were implemented:<br>• Exploration<br>• Vertigo<br>• Free fall<br>• Incarnation | NA | 18 sessions over 6 weeks (3 times a week with 25 min per session) | NA | NA |
| Cieslik, 2023 | IVR as support for the mental health of elderly women | Effectiveness of an IVR garden aesthetics, relaxation with elements of Erickson psychotherapy in alleviating depression and anxiety Theory: Erickson's Psychotherapy | VR goggles Two controllers VR stimulation elements:<br>• Aspects of Erickson's psychotherapy<br>• Relaxing music<br>• Cognitive stimulation (colouring mandalas using controllers)<br>• Garden environment | Physiotherapist to conduct general fitness training session The music was co-composed by a music therapist and a music composer | 8 sessions over 4 consecutive weeks | Tailored relaxation music composed specifically for this study by music therapists and composer | 30 participants were allocated into the VR group and completed the VR sessions |
| Mazurek, 2023 | IVR and Orthopedic Rehabilitation | The efficacy of VR therapy in reducing depression, anxiety, and stress among older adults following arthroplasty surgery Theory: Erickson Psychotherapy principles | VRTierOne device (VR HTC VIVEW goggles + 2controllers) VR stimulation elements:<br>• Aspects of Erickson's psychotherapy<br>• Relaxing music<br>• Cognitive stimulation (colouring mandalas using controllers)<br>• Green garden | NA | 8 sessions over 4 weeks (twice per week) | NA | All completed the intervention |

*(Continued)*

**Table 2.** (Continued)

| First author, year | Brief name | Goals & theory | Material & procedure | Professional input | how much | Tailoring | Actual usage |
|---|---|---|---|---|---|---|---|
| Szczepańska-Gieracha, 2021 | Virtual therapeutic garden | Effect of IVR in the elderly for improving depressive symptoms. Theory: Erickson psychotherapy, positive psychology | VRTierOne (Stolgraf), VR goggles, controllers, and head-mounted display VR therapeutic garden (garden of revival, symbolize patient health, the garden will turn from grey to more colourful) | The content IVR was developed by a certified psychotherapy therapist | Eight 20-min long VR sessions over four weeks (twice per week) | Incorporate breathing exercises, elements of mindfulness training and hypnotic suggestions to enrich the therapeutic effect | 13 participants were allocated into the VR group and completed the VR sessions |
| Tominari, 2021 | Reminiscence therapy for older adults with dementia | IVR reminiscence versus conventional reminiscence on cognitive functions, subjective well-being, and ADL Theory: NA | iPad & 360-degree panoramic displays Papers Tailored Japanese reminiscence content (historical scenes) | A certificated nurse trained in reminiscence therapy conducted all 8 reminiscence sessions | Eight one-on-one 30–45-min VR-based reminiscence therapy over 8 weeks | Tailored Japanese content Selected photos of material and cultural artifacts corresponding to the childhoods of participants | NA |
| Yuan, 2022 | Restorative effect of VR forests | the restorative effects of IVR forest experiences on elderly people during the COVID-19 lockdown Theory: Forest therapy Attention restoration theory Stress recovery theory | VR forest scenarios head-mounted display (VR SHINECON AIO5), This device contained gyroscopes, nine axis sensors, headphones, and other auxiliary accessories | NA | 3 VR forest experience sessions over 3 days | Tailored Chinese scenery In the VR environment, Black Valley National Forest Park, was presented. | NA |

three-dimensional photos of famous tourist hotspots in the city where participants lived [39], historical pictures from participants' youth provided by local museums [36], and images from a national forest park in the participants' country [37]. Additionally, some interventions included tailor-made relaxation music composed by a music therapist [41], incorporated breathing exercise [43], or task of coloring mandalas using controllers [41,42]. Participant positioning during IVR interventions varied across studies. Three studies explicitly reported a seated position to ensure comfort and minimize adverse effects [35,37,42], one reported using both sitting and standing positions [41], and one employed a supine (lying) position [38]. Another study involved multi-sensory stimulation with participants in a prone position using a harness system [40].

## Depressive symptoms

Five studies examined the effects of IVR interventions, predominantly focusing on garden relaxation, except for one study that utilized multi-sensory stimulation [40]. Depressive symptoms were assessed using the Geriatric Depression Scale [38,41,43], Yesavage scale [40], or Hospital Anxiety and Depression Scale (HADS) -depression subscale [42]. As shown in Fig 4, participants who received IVR interventions experienced a significant reduction in depressive symptoms compared to the control group [n = 5; SMD = -0.83, 95% CI (-1.05, -0.60), $I^2$ = 21%, p < .001]. The control group received usual care [40,42,43], group relaxation and psychoeducation [41], or Schultz's Autogenic Training [38]. Certainty of the evidence was rated as moderate, downgraded by one level for risk of bias due to lack of blinding of participants and personnel in all studies and unclear allocation concealment in two studies [40,43], which could increase the risk of performance and selection bias for subjective outcomes of self-reported depressive symptoms.

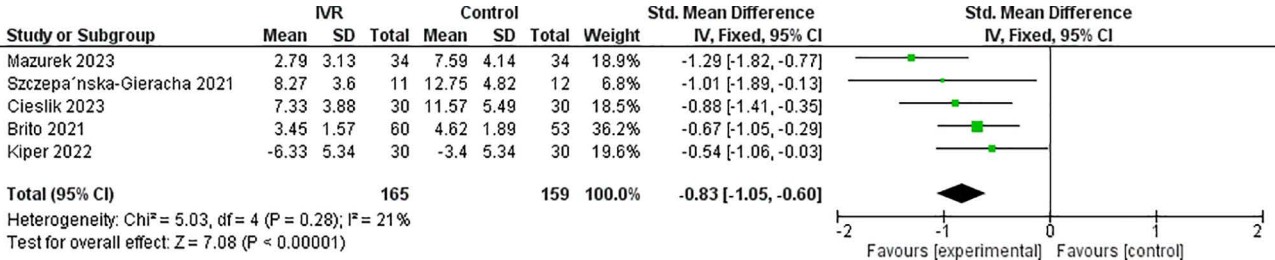

**Fig 4. Effect on depressive symptoms.**

## Anxiety

Anxiety was assessed in six studies using the HADS-anxiety subscale [38,41–43], General health questionnaire anxiety subscale [40], or State-Trait Anxiety Inventory [35]. As shown in Fig 5, pooled data from five of these studies showed a significant reduction in anxiety [n = 5, SMD = -0.77, 95% CI (-1.32, -0.22), $I^2$ = 70%, $p$ = .006]. One study, which used multi-sensory stimulation and reported a significant reduction in anxiety levels, was not included in the pooled analysis due to the lack of reported mean and standard deviation values [40]. A sensitivity analysis excluding one study that compared IVR with Schultz's Autogenic Training [38], a desensitization technique, showed a large, statistically significant reduction in anxiety with minimal heterogeneity [n = 4, SMD = -1.01, 95% CI (-1.35, -0.68), $I^2$ = 1%, $p$ < .001]. Certainty of the evidence was rated as moderate, downgraded by one level for risk of bias due to lack of blinding of participants and personnel in all studies and unclear allocation concealment in two studies [43], which could increase the risk of performance and selection bias for subjective outcomes of self-reported depressive symptoms.

## Stress

Three studies evaluated IVR nature exposure (garden or forest) to reduce stress among post–joint arthroplasty patients [42], older women with depression [43], or institutionalized older adults during the COVID-19 lockdown [37]. All studies compared IVR psychological intervention with standard care regarding its effect on stress, measured by Perceived Stress Scale [42], Perception of Stress Questionnaire [43], six-item Restorative Outcome Scale [37]. Meta-analysis was not feasible because only two studies reported means and standard deviations with Hedges'g = 1.08 [42] and 1.15 [43]. All three studies (3/3; 100%) reported statistically significant reductions in stress in the IVR groups compared with controls. Evidence consistently favors IVR psychological intervention for stress reduction across diverse older adult populations, but heterogeneity of measures and incomplete reporting limit precision and preclude estimation of effect magnitude.

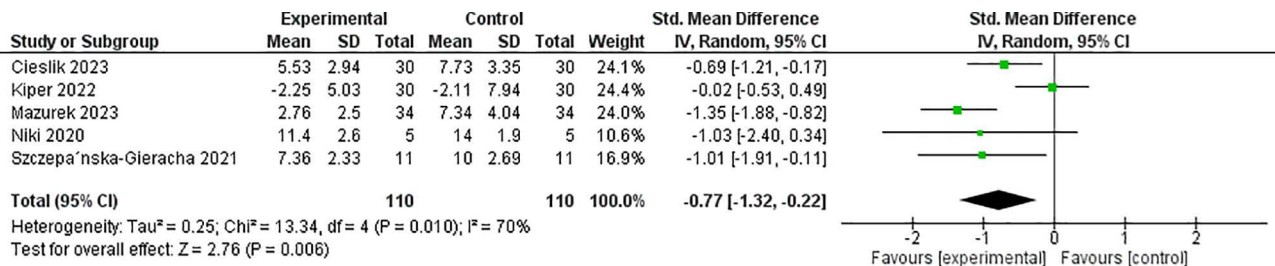

**Fig 5. Effect on anxiety.**

### Psychological wellbeing

Overall psychological well-being was assessed in three studies using the RYFF Wellbeing Scales [34,40], and the PGC Morale Scale [36]. As shown in Fig 6, pooled data indicated no significant improvement in psychological well-being [n = 3, SMD = 0.45, 95% CI (-0.57, 1.46), $I^2$ = 91%, p = 0.39]. Given the unexplainable heterogeneity, the inclusion of only three studies, and the high inconsistency of the results, along with limitations related to risk of bias, a low certainty of evidence is considered appropriate.

### Affect

Two studies examined IVR psychological intervention among institutionalized older adults during the COVID-19 lockdown [37] and community-residing older adults [39]. One evaluated IVR-based cognitive stimulation versus paper-and-pencil cognitive stimulation [39]; the other examined an IVR forest/nature exposure intervention versus usual care [37].

Both studies measured changes in affect using the Positive and Negative Affect Schedule [37,39]. Both studies (2/2; 100%) reported statistically significant reductions in negative affect and increases in positive affect for IVR interventions compared with their respective controls. The consistent direction of effect suggests IVR-based cognitive and nature-exposure interventions may improve affect in older adults. However, the evidence base is limited (two studies), interventions and comparators differ, and data reporting did not permit estimation of a pooled effect size.

### Adverse events

Three studies evaluated adverse events associated with IVR use; only one detailed participant positioning. In the study employing a seated position, nausea (n = 1), tiredness (n = 2), and headache (n = 1) were reported [35], whereas another study reported severe simulator sickness in 1.4% of participants [39]. Another study reported mild nausea and vestibular/oculomotor disturbances in 5–20% of participants [34].

### Discussion

To our knowledge, this systematic review and meta-analysis is the first to aggregate empirical data from RCTs to evaluate the effect of IVR-based psychological intervention on the psychological well-being of older adults. Our meta-analysis suggests that IVR-based interventions lead to significant improvements in depressive symptoms and anxiety with moderate evidence certainty. Additionally, a synthesis without meta-analysis indicated that IVR-based interventions significantly improved stress levels and affect in older adults. However, these interventions showed inconclusive effects on overall psychological well-being. The synthesis of intervention engagement revealed a high completion rate for the IVR sessions, indicating good usability.

The effectiveness of IVR-based psychological interventions in reducing depressive symptoms among older adults is noteworthy. This finding aligns with previous literature that has demonstrated the effectiveness of traditional, non–technology-based psychological interventions in alleviating depressive symptoms in this population [44,45].

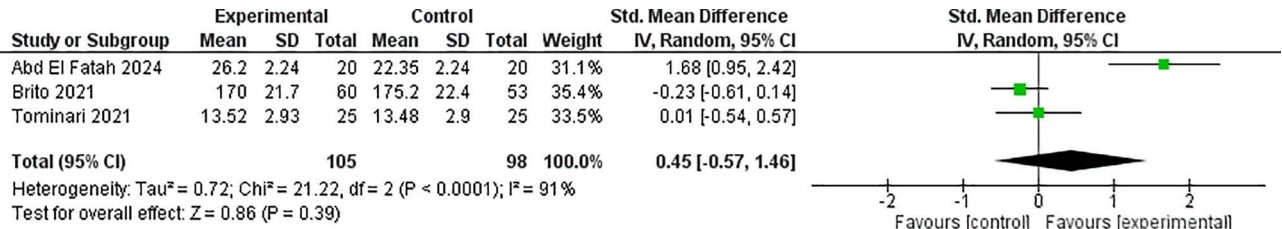

**Fig 6. Effect on psychological wellbeing.**

Additionally, the pooled studies predominantly utilized garden therapy as key therapeutic element to achieve these benefits, and the advantages of garden or nature exposure have been widely documented across diverse contexts, including wilderness areas, urban green spaces, and gardens [46,47]. Moreover, IVR technology offers distinct advantages in terms of tactile engagement of the garden and its capacity to replicate the complex interconnectedness of garden/nature environments. It accomplishes this by manipulating surroundings and sensory modalities to create immersive experiences that closely resemble real-world environments [47]. Moreover, IVR based garden exposure is stated to generate physiological effects of suppressing sympathetic nervous activity, which may also explain the improvement in depressive symptoms and potential reduction in stress level [48]. Although proposed physiological mechanisms sound plausible, evidence in older adults is limited as many studies use small samples, heterogeneous protocols and measures, unstandardized dosage, mechanistic claims should be considered provisional [22]. IVR's distinctive capability to create vivid and interactive experiences with garden environments presents an opportunity to deliver interventions to older adults with mobility difficulties and limited access to natural environments [46]. For these individuals, while engaging with nature has been theoretically therapeutic but not feasible, interacting with "virtual nature" has now also been shown to be therapeutic.

The finding indicated that IVR intervention reduced anxiety level, which is contributed by using garden therapy and reminiscence therapy. The garden IVR therapy is believed to evoke memories of viewing certain landscape images, thereby inducing a sense of comfort and decreasing the heart rate [41,49]. Reminiscence intervention is a narrative and reflective process that helps older adults to recall of past experiences, emotions, and thoughts to solicit positive feelings, coping mechanism and adaptation [50]. IVR technology hold unique strength that during reminiscence sessions, tangible prompts such as historical environment, old photographs, or videos can be used to immerse older adults by manipulate surroundings and sensory modalities to evoke past memories and stimulate conversations [51]. Through sharing retrospective accounts of one's past, reminiscence-based intervention aims to enhance positive feelings, personal identity and buffer against depressive symptoms.

The impact of IVR on overall psychological wellbeing was inconclusive, with high heterogeneity across studies. This may be because the applied interventions did not fully address the multifaceted nature of psychological wellbeing faced by older adults. Improving psychological wellbeing requires improvement in meaning and self-realization, life satisfaction, perception towards aging [52], as well as having a key influential role towards physical health [53]. Effective interventions would need to go beyond psychological interventions to integrate elements such as physical exercises, social skills training, and functional capacity enhancement to holistically address psychological wellbeing. This underscores the necessity of tailoring the IVR intervention designs tailored to the health needs of older adults as their health and wellbeing varies substantially. Additional exploration into participant characteristics or intervention components contributing to variability in outcomes would enhance future research clarity.

Moreover, a deeper examination IVR modalities in the included studies highlights several strengths over non-immersive or non-technology-based methods: high presence and interactivity that foster engagement; capacity for personalization (e.g., incorporating personal photos, hometown imagery, and tailored relaxation music [34–36,41] and a confidential, controllable environment that supports self-disclosure while enabling remote monitoring, feedback, and scalable delivery. By supporting uninterrupted immersion and aligning content with older adults' life experiences, IVR may promote sustained participation in positive, meaningful activities and yield more durable effects [18]. IVR should be viewed as a tool that complements, not replaces, the therapeutic encounter. Mental health professionals remain essential for assessment and safety, tailoring content to patients' psychosocial context, and ongoing adjustment of treatment plans, thereby preserving therapeutic rapport and maximizing applicability and impact [11].

Two concerns regarding its implementation warrant attention. First, adverse events associated with IVR have been insufficiently assessed, despite their importance to research and clinical practice. High completion rates across studies, coupled with reports that cybersickness is generally mild or tolerable, indicate good usability and acceptability. Seated use and teleportation locomotion [54], together with contemporary advances in display latency, refresh rates, optics, and

tracking [55], may substantially mitigate cybersickness; however, this requires further investigation among older adults. Second, feasibility issues related to cost, technical literacy, and infrastructure in healthcare facilities and setting, critical for real-world scale-up, also merit further study.

## Implications

Several practice implications warrant consideration. First, considering that IVR-based psychological interventions are effective in alleviating mental health symptoms, increasing accessibility, and engaging older adults, it may serve as a viable alternative to traditional yet effective psychological interventions that are more labor-intensive, such as conventional horticultural therapy or acupressure, for promoting mental health among older individuals [56–58]. Second, it is advisable to exercise selectivity by assigning patients to IVR psychological interventions. Patients who prefer not to use technology experience adverse effects from IVR or have either minimal or severe psychological conditions may benefit more from alternative interventions. Given the variability in participant positioning across studies and recent findings by [59] showing that seated IVR use significantly reduces cybersickness symptoms such as eye strain, headaches, and discomfort, future interventions should clearly report and justify participant positioning to optimize comfort, minimize adverse effects, and support adherence. Finally, real-world uptake depends on practical considerations, including cost (e.g., start-up expenditures for hardware, sanitation accessories, and software licensing, as well as ongoing maintenance and support), technical literacy, facility-level infrastructure (e.g., reliable broadband; and integration with existing health information systems, with adherence to privacy and cybersecurity standards), and more robust data on the prevalence and severity of adverse events. Transparent reporting and proactive planning for these factors will enhance the translational relevance and practical feasibility of IVR interventions in routine care.

Furthermore, several implications for future research are noteworthy. Further studies should also compare the acceptance and cost-effectiveness between the IVR-enhanced interventions and the interventions delivered in a conventional way. A dedicated review on tailoring strategies in IVR would be valuable once a larger body of tailored IVR intervention studies becomes available to identify the most effective intervention types or tailoring features. Studies may gain from integrating IVR psychological interventions with other components tailored to participants' characteristics and research objectives. For instance, if participants have both mental health issues and cognitive impairments, IVR psychological interventions could include activities such as virtual garden sightseeing and painting for cognitive stimulation, thereby enhancing both psychological and cognitive outcomes. Strategies such as enhancing IVR interactivity through sensors and controllers, employing adaptive algorithms for personalization, and incorporating participants' personal narratives or memories to create immersive experiences may help achieve outcomes comparable to those of traditional psychological interventions. Lastly, future research may engage stakeholders to address implementation barriers, including cost, technical literacy, and infrastructure within care facilities that are critical to real-world adoption.

## Limitations

Several limitations warrant consideration. Many included studies enrolled small samples, limiting generalizability. The number of eligible trials was insufficient to conduct meta-regression to identify the most effective intervention types or tailoring features. Substantial heterogeneity in psychological well-being outcomes could not be resolved through sensitivity analyses because of the limited study number, necessitating cautious interpretation. Funnel-plot asymmetry was not assessed because fewer than 10 studies were pooled, which precludes distinguishing chance from true asymmetry; nonetheless, publication bias was evaluated by checking for selective reporting against trial registrations and published protocols. Across the included RCTs, inadequate allocation concealment and lack of blinding of participants and personnel increased risk of bias, leading to downgrading the certainty of evidence to low or moderate. Taking together, these constraints temper confidence in the observed effects and their applicability; future work should prioritize adequately powered, methodologically rigorous trials to clarify effectiveness, reduce heterogeneity, and generate decision-ready evidence for practice and policy.

## Conclusion

IVR-based psychological interventions have showed potential effectiveness in alleviating depressive symptoms, anxiety, stress, and improving affect among adults with low- to moderate-quality evidence. The immersiveness, interactivity, and individual tailoring of the included IVR interventions, along with their incorporation of evidence-based psychological therapies (e.g., nature exposure, reminiscence), may have contributed to the effectiveness. However, given study limitations, the lack of standardized intervention protocols and dosing, and inconclusive effects on psychological well-being, further research is warranted. Rigorous, adequately powered trials with evidence-based content, standardized dosage, and longer follow-up are essential to substantiate these promising findings.

## Supporting information

**S1 File. Includes 1) MeSH terms and search strategies for each database, and 2) a table detailing the screening process.**
(DOCX)

**S1 Checklist. PRISMA checklist.**
(DOCX)

## Author contributions

**Conceptualization:** Jing Jing Su, Chi-Keung Chan, Ladislav Batalik, Rick Yiu Cho Kwan.

**Formal analysis:** Jing Jing Su, Wai Chung Chung, Chen Lei.

**Methodology:** Jing Jing Su, Ladislav Batalik, Rick Yiu Cho Kwan.

**Supervision:** Rick Yiu Cho Kwan.

**Writing – original draft:** Jing Jing Su.

**Writing – review & editing:** Jing Jing Su, Chi-Keung Chan, Ladislav Batalik, Rick Yiu Cho Kwan.

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
