## [Decision Letter · Decision Letter 0]

27 Aug 2025

Response to Reviewers
Revised Manuscript with Track Changes
Manuscript
**Journal Requirements:**
**Additional Editor Comments (if provided):**

Major revisions:

1. Fix internal inconsistencies and reporting errors:

- The Results text states “after removing 686 duplicates,” while the PRISMA diagram shows “Duplicates n = 544.”. Please reconcile counts across text/figure and ensure every box total is auditable from your search logs.

- The abstract reports SMD = 0.07; 95% CI (−0.99, −0.44), which conflicts with the main text/forest plot (SMD ≈ −0.77; 95% CI (−1.32, −0.22)). Please correct the sign and values and harmonize across Abstract, Results, and Figures.

2. Clarify and justify eligibility decisions.

- You exclude IVR physical and cognitive‑only training, but include trials where IVR was delivered alongside sensorimotor or rehabilitation components when psychological outcomes were primary or secondary. Please explain this rationale, define what you considered a “psychological” IVR intervention, and justify inclusion of mixed interventions.

- Please define “immersive IVR” operationally (hardware, FoV, interaction) to distinguish from non‑immersive VR.

3. For outcomes not meta‑analyzed (e.g., stress, affect), structure the narrative per SWiM guidance (what was grouped, how direction/size of effects were decided, and certainty considerations).

**Reviewers' Comments:**

**Comments to the Author**

1. Does this manuscript meet PLOS Digital Health’s publication criteria?

Reviewer #1: Yes

Reviewer #2: Yes

Reviewer #3: Yes

Reviewer #4: Yes

2. Has the statistical analysis been performed appropriately and rigorously?

Reviewer #1: Yes

Reviewer #2: N/A

Reviewer #3: Yes

Reviewer #4: Yes

3. Have the authors made all data underlying the findings in their manuscript fully available (please refer to the Data Availability Statement at the start of the manuscript PDF file)?

Reviewer #1: Yes

Reviewer #2: Yes

Reviewer #3: Yes

Reviewer #4: Yes

4. Is the manuscript presented in an intelligible fashion and written in standard English?

Reviewer #1: Yes

Reviewer #2: Yes

Reviewer #3: Yes

Reviewer #4: Yes

Reviewer #1: Immersive virtual reality (IVR) is an emerging therapeutic modality that engages older

adults in psychological therapeutically orientated activities developed to improve their

psychological well-being. This systematic review aims to investigate the effects of IVR

psychological intervention on psychological symptoms and well-being. A systematic

review and meta-analysis was conducted following the Cochrane Handbook for

Systematic Reviews of Interventions. Six databases were searched, including Embase,

PubMed, Web of Science, Scopus, CINAHL, and PsycINFO, covering the period from

2010 to December 2024. RevMan 5.3 was utilized for meta-analysis, and the Cochrane

Risk of Bias tool was employed for quality assessment. Ten randomized controlled

trials of 746 older adults were included. The IVR interventions employed reminiscence

(40%), garden/forest therapy (40%), cognitive stimulation (10%), and sensor motor

rehabilitation (10%). Data pooling suggested that IVR interventions have significantly

reduced depressive symptoms [n=5; SMD=-0.83, 95%CI (-1.05, -0.60), I2=21%,

p<.001]; anxiety [n=5, SMD=0.07, 95% CI (-0.99, -0.44), I2=70%, p<.001]. Synthesis

without meta-analysis showed significant improvement in stress level and affect. IVRbased interventions could be an alternative method for alleviating the psychological

symptoms of older adults.

Author summary

This study investigated the effects of immersive virtual reality (IVR)-based

psychological interventions in alleviating psychological symptoms and enhancing wellbeing among older adults. The novelty of this research lies in its pooling of results from

updated randomized controlled trials that exclusively examine the immersive format of

VR-delivered psychological interventions. Previous studies often combined varying

levels of VR immersiveness or mixed different types of IVR-based interventions (e.g.,

exercise, cognitive assessment, psychological), resulting in high heterogeneity that

impeded definitive conclusions. The impact of this study is demonstrated by employing

advanced immersive technology to create vivid and interactive experiences (e.g.,

garden environments) that achieve therapeutic effects. The meta-analysis results

revealed that IVR-based psychological interventions significantly reduced depressive

symptoms and anxiety among older adults, indicating the high potential of this modality

for improving psychological outcomes, particularly for those with mobility limitations.

can be published in the journal

Reviewer #2: It is a very important and interesting article, thank you for your efforts. I think it will be a strong reference for many future related works. So, I suggest to make two points (Optional):

- Regarding tailored or personalized treatment especially in this context, can you suggest how to determine the most tailored or personalized intervention?

- List most effective interventions according to your analysis, to be used as a reference in future works.

Reviewer #3: Thank you for the opportunity to review this timely and relevant manuscript on immersive virtual reality (IVR)-based psychological interventions for older adults. The topic is essential, and the systematic review is methodologically well-grounded, with clear eligibility criteria, a comprehensive search strategy, and a balanced use of meta-analysis and narrative synthesis. The manuscript also does a good job of linking its findings to prior literature and highlighting their practical implications.

That said, there are several areas where the paper could be strengthened before publication. These include refining the introduction to specify the mental health challenges in older adults clearly and explicitly link them to the rationale for IVR, improving the flow and avoiding repetition in the discussion, clarifying specific methodological details (e.g., rationale for exclusion criteria, full author names in the study selection section, consistent use of effect size terminology), and presenting a more balanced interpretation of findings by giving equal weight to limitations and potential biases. Statistical reporting can also be streamlined by using confidence intervals instead of both CIs and p-values in the abstract and results.

In the discussion and conclusion, I encourage the authors to temper overly optimistic language, especially where evidence is limited or heterogeneity is high, and to address potential real-world barriers to implementation such as cost, technical literacy, and care-setting infrastructure. Finally, while the conclusion appropriately calls for larger and more rigorous trials, it would be strengthened by also recommending more diverse study populations and standardized outcome measures to improve comparability across studies.

I have attached a detailed review of the manuscript, including specific section-by-section feedback and suggested revisions, for your consideration.

Reviewer #4: General Impressions and comments:

I am really honored to review this significant and remarkable work. Efforts and collaboration by all authors deserve many thanks and appreciation, as the manuscript demonstrates a beacon modality for all else.

The study implemented a robust methodology to review systematically the effectiveness of a critical concerns entitled "Utilities and Interventions of Virtual Technology". Description by the authors on the research methods, strategies, and implementation are lucid clear. Kudos to that.

The study and its approach represent a significant and timely matter, and demonstrate its eligibility for a broader readerships and scientific community.

One of the strategic implementation I would like to thank authors on is the way of handling disagreement. The authors have ensured a multi-pronged approach and a well-defined implementation strategy, with exemplary impact and effectiveness, ensured a high-quality manuscript, which greatly enhances the robustness and strength of the methodology employed in this systematic review and meta-analysis. For instance, the authors considered discussion among them when it comes to disagreement on study selection. This precise excellent strategy implemented are optimum and highly effective for a key aspect associated with the overall manuscript and its quality, preparation, and development. The collaborative discussions among the authors regarding study selection demonstrate an exemplary commitment to transparency and rigor in the review process, contributing significantly to the overall quality and preparation of the manuscript. In contrast, the authors considered a third author to solve disagreement on interpretation of the extracted data among 2 authors. Such a tailored policy are conclusive and categorical for a particular aspect that doesn't effect the overall quality, preparation and development of the manuscript, particularly when it comes to personal perspectives and preferences to elucidate findings from a similar data extracted. While the involvement of a third author to resolve disagreements on the interpretation of extracted data is commendable and reflects a tailored approach to ensuring comprehensive analysis, it is important to note that this strategy does not detract from the overall quality of the manuscript. Instead, it reinforces the integrity of the findings by addressing personal perspectives and preferences.

Overall, these strategies have positively influenced the manuscript’s development, ensuring that the conclusions drawn are well-founded and reflective of a thorough and balanced assessment of the evidence.

The manuscript comprehensively discuss the entire context of the research title and the overall landscape aspects, including alignment to findings from previous literature and limitations to consider in the future directions of upcoming work.

The authors demonstrated an efficient capacity to deliver the results/findings of this research in well-written statement under the conclusion section.

Minor Revision Feedback:

There is a kind of meaningless iteration/repeat of the conjunctive adverb "However" in the last paragraph of the introduction section (However, the current uncertainty..../However, existing reviews...).

Please avoid such kind of repeated writing, particularly in introducing or contrasting 2 relevant statements using the same conjunction term. You may consider synonym term instead (E.x: Nonetheless, nevertheless...etc)

There is mismatching in data. In the Results section, the content texted under the subsection "search outcome" indicate the removal of 686 duplicates. However, duplicate n was equal to 544 in the figure of PRISMA flow diagram. Kindly modify with the correct matched data.

Overall Comment:

I highly recommend the publication of this remarkable piece of evidence and science, as it demonstrates its uniqueness to be the first of its kind.

**Do you want your identity to be public for this peer review?** For information about this choice, including consent withdrawal, please see our Privacy Policy

Reviewer #1: No

Reviewer #2: **Yes: ** Jehad Omar Abualrob

Reviewer #3: **Yes: ** Victor Ifechukwude Agboli

Reviewer #4: **Yes: ** Omar Abdulkarim Saeed Alhammadi‬‏

**Figure resubmission:**

**Reproducibility:** To enhance the reproducibility of your results, we recommend that authors of applicable studies deposit laboratory protocols in protocols.io, where a protocol can be assigned its own identifier (DOI) such that it can be cited independently in the future. Additionally, PLOS ONE offers an option to publish peer-reviewed clinical study protocols. Read more information on sharing protocols at https://plos.org/protocols?utm_medium=editorial-email&utm_source=authorletters&utm_campaign=protocols

---

## [Decision Letter · Decision Letter 1]

13 Nov 2025

Immersive Virtual reality-based intervention for psychological wellbeing among older adults: A systematic review and meta-analysis

PDIG-D-25-00484R1

Dear Prof Kwan,

We are pleased to inform you that your manuscript 'Immersive Virtual reality-based intervention for psychological wellbeing among older adults: A systematic review and meta-analysis' has been provisionally accepted for publication in PLOS Digital Health.

Best regards,

Phat Kim Huynh, Ph.D.

Guest Editor

PLOS Digital Health

**Additional Editor Comments (if provided):**

**Reviewer Comments (if any, and for reference):**

Reviewer's Responses to Questions

**Comments to the Author**

Reviewer #2: All comments have been addressed

Reviewer #3: All comments have been addressed

Reviewer #5: All comments have been addressed

Reviewer #6: All comments have been addressed

publication criteria?

Reviewer #2: Yes

Reviewer #3: Yes

Reviewer #5: Yes

Reviewer #6: Yes

3. Has the statistical analysis been performed appropriately and rigorously?

Reviewer #2: Yes

Reviewer #3: Yes

Reviewer #5: Yes

Reviewer #6: Yes

4. Have the authors made all data underlying the findings in their manuscript fully available (please refer to the Data Availability Statement at the start of the manuscript PDF file)?

Reviewer #2: Yes

Reviewer #3: Yes

Reviewer #5: Yes

Reviewer #6: Yes

5. Is the manuscript presented in an intelligible fashion and written in standard English?

Reviewer #2: Yes

Reviewer #3: Yes

Reviewer #5: Yes

Reviewer #6: Yes

Reviewer #2: Thank you

Reviewer #3: Thank you for your thorough and thoughtful revisions. You have addressed all my previous comments with substantial improvements in conceptual clarity, methodological transparency, and interpretive balance. The revised manuscript now presents a clear, well-organized synthesis of evidence on immersive virtual reality (IVR)–based psychological interventions for older adults, with consistent statistical reporting, clarified eligibility criteria, and explicit adherence to PRISMA, SWiM, and GRADE guidelines. The introduction effectively connects mental health challenges in aging to the rationale for IVR, and the discussion now offers a balanced interpretation that acknowledges heterogeneity, study limitations, and implementation barriers such as cost and technical literacy. Aside from minor stylistic refinements for conciseness, the manuscript is strong, polished, and ready for publication.

Reviewer #5: Dear Authors,

The manuscript presents a well-structured and methodologically rigorous systematic review and meta-analysis on the effectiveness of immersive virtual reality (IVR) psychological interventions in improving mental health outcomes among older adults. Major strengths include the precise operational definition of immersive VR, clear differentiation between immersive and non-immersive interventions, transparent use of the RoB2 and GRADE frameworks, and comprehensive reporting of both meta-analytic and narrative results. The work offers an important contribution to the field of gerontechnology and digital therapeutics by consolidating current evidence on IVR’s impact on depression, anxiety, and stress reduction.

Despite its overall quality, two aspects could be refined before publication. First, the discussion section should slightly adjust its tone: rather than stating that IVR interventions “demonstrated effectiveness,” it would be more accurate to use balanced phrasing such as “showed potential effectiveness,” reflecting the moderate certainty of evidence and the limited sample sizes of included trials. Second, the section on clinical applicability would benefit from a clearer presentation of implementation aspects—particularly quantitative information on adverse-event frequency and concise commentary on real-world barriers such as cost, technical literacy, and infrastructure requirements in care facilities. These additions would strengthen the translational and practical value of the conclusions.

Overall, the manuscript demonstrates scientific rigor, conceptual clarity, and high relevance for interdisciplinary readership. The article can be considered suitable for acceptance and publication.

Reviewer #6: Thanks for the opportunity to review the manuscript "Immersive Virtual reality-based intervention for psychological wellbeing among older adults: A systematic review and meta-analysis" (PDIG-D-25-00484R1).

This paper addresses an important topic and explores a significant question regarding the intersection of technology and mental health in an aging population.

It is evident that the authors have carefully considered the feedback from previous reviewers, and their responses are thoughtful and appropriate. In particular, I appreciate the addition of a more detailed description of immersive virtual reality (IVR) and a clear definition of psychological intervention. These enhancements not only clarify the concepts for readers but also strengthen the overall framework of the study.

Thank you once again for the opportunity to engage with this important work.

**Do you want your identity to be public for this peer review?** For information about this choice, including consent withdrawal, please see our Privacy Policy

Reviewer #2: **Yes: ** Jehad Omar Abualrob

Reviewer #3: **Yes: ** Victor Ifechukwude Agboli

Reviewer #5: No

Reviewer #6: **Yes: ** Cho Lee Wong
